# Self-assembly of tessellated tissue sheets by expansion and collision

Matthew A. Heinrich[1,8], Ricard Alert [2,3,4,5,8], Abraham E. Wolf [6,8], Andrej Košmrlj [1,7✉] & Daniel J. Cohen[1,6✉]

Tissues do not exist in isolation—they interact with other tissues within and across organs. While cell-cell interactions have been intensely investigated, less is known about tissue-tissue interactions. Here, we studied collisions between monolayer tissues with different geometries, cell densities, and cell types. First, we determine rules for tissue shape changes during binary collisions and describe complex cell migration at tri-tissue boundaries. Next, we propose that genetically identical tissues displace each other based on pressure gradients, which are directly linked to gradients in cell density. We present a physical model of tissue interactions that allows us to estimate the bulk modulus of the tissues from collision dynamics. Finally, we introduce TissEllate, a design tool for self-assembling complex tessellations from arrays of many tissues, and we use cell sheet engineering techniques to transfer these composite tissues like cellular films. Overall, our work provides insight into the mechanics of tissue collisions, harnessing them to engineer tissue composites as designable living materials.

[1] Department of Mechanical and Aerospace Engineering, Princeton University, Princeton, NJ 08544, USA. [2] Princeton Center for Theoretical Science, Princeton University, Princeton, NJ 08544, USA. [3] Lewis-Sigler Institute for Integrative Genomics, Princeton University, Princeton, NJ 08544, USA. [4] Max Planck Institute for the Physics of Complex Systems, Nöthnitzerst. 38, 01187 Dresden, Germany. [5] Center for Systems Biology Dresden, Pfotenhauerst. 108, 01307 Dresden, Germany. [6] Department of Chemical and Biological Engineering, Princeton University, Princeton, NJ 08544, USA. [7] Princeton Institute of Materials, Princeton University, Princeton, NJ 08544, USA. [8] These authors contributed equally: Matthew A. Heinrich, Ricard Alert, Abraham E. Wolf. ✉email: andrej@princeton.edu; danielcohen@princeton.edu

A biological tissue is a cellular community or, as Virchow wrote in the 19th century[1], "a cell state in which every cell is a citizen". This concept is increasingly apropos as interdisciplinary research pushes deep into the coordinated cell behaviors underlying even "simple" tissues. Indeed, cell–cell interactions give rise to behaviors such as contact inhibition[2–4], collective cell migration[5,6], and cell-cycle regulation[7–10], which underlie physiological functions such as tissue development and healing[11,12], organ size control[13,14], morphogenetic patterning[15], and even pathological processes such as tumor invasion[16,17].

In places where tissues meet, the resulting tissue is a living composite material whose properties depend on its constituent tissues. In particular, tissue–tissue interfaces underlie both biological processes such as organ separation and compartmentalization[18,19], as well as biomedical applications such as tissue-mimetic materials[20–22] and engineered tissue constructs[23–25]. Thus, recent research has focused on the formation and dynamics of tissue–tissue boundaries. For instance, the interplay between repulsive Eph/ephrin and adhesive cadherin cell–cell interactions regulate tissue boundary roughness, stability, and cell fate[26–30]. Furthermore, colliding monolayers with differences in Ras gene expression were able to displace one another[31,32], while epithelial tissue boundaries were found to induce waves of cell deformation and traction long after the tissues had collided[33].

Here, we sought to broadly investigate questions such as: how do tissues of different shapes interact with each other, and what happens when many tissues simultaneously interact? Our goal was to then develop these fundamental concepts into broad "design principles" for assembling composite tissues in a controlled way. Specifically, we sought to harness mechanical tissue interactions in the context of both tissue biophysics and cell-sheet engineering, where the latter aims to produce and harvest intact cell monolayers to create scaffold-free, high-density tissues[24]. Such cell sheets are typically produced by allowing cells to come to confluence within a stencil or patterned substrate to form a monolayer with a desired geometry[34,35]. Alternatively, tissue composites can be produced by harvesting arrays of small, autologous epidermal patches to fuse with time over burn wounds[36].

To facilitate this work, we developed a rapid tissue patterning and arraying approach based on low-cost razor writing of silicone micro-stencils[10]. Using this technique, we created arrays of individual epithelial monolayers and then allowed them to grow out and collide, fuse at the interfaces, and ultimately self-assemble into tessellated patterns. During this process, we performed live imaging as these tissue arrays self-assembled into patterns over 2–3 days, which we were able to predict by extending our earlier model of tissue expansion[10] to account for multi-tissue interactions. We then characterized the dynamics of the boundary in collisions of tissues with different sizes and cell densities. Moreover, we proposed a physical model for understanding the resulting boundary motion and extracting tissue mechanical properties from it. We next introduced a design framework for the systematic assembly of many-tissue composites (3 cell types and 30+ tissues), and investigated more complex cases such as the singular dynamics of tri-tissue junctions. Finally, we introduced heterotypic tissue collisions as an area for future development, which enabled additional features such as tissue engulfment.

## Results

**Collisions between archetypal tissue pairs.** Our first experiment aimed at uncovering the basic rules governing tissue collisions, with the goal to understand and then fully predict the shape and

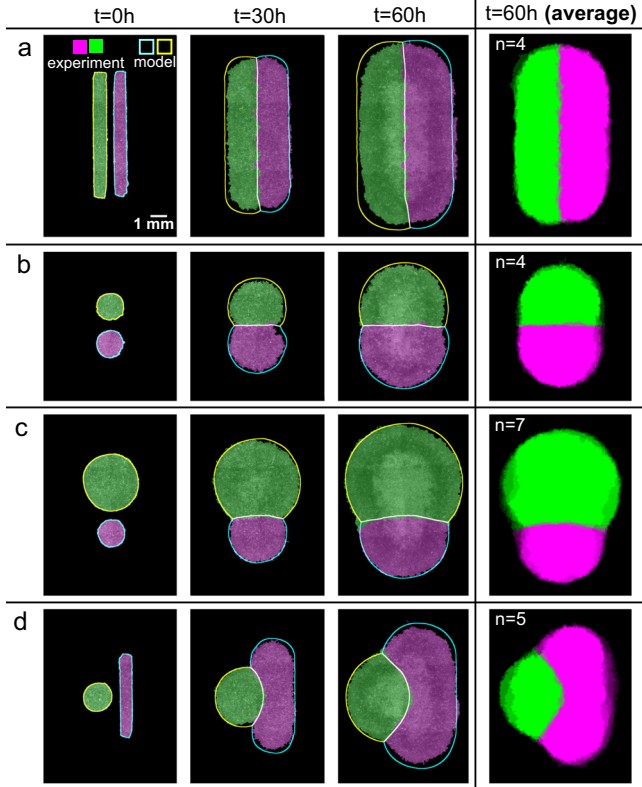

**Fig. 1 The shapes of colliding tissues are stereotyped and predictable. a–d** Archetypal collision experiments (solid) and simulations (outline) for equally sized rectangles (**a**), equally sized circles (**b**), size-mismatched circles (**c**), and circle-rectangle pairs (**d**). Averages over several tissues at 60 h are shown in the rightmost panels ($n = 4$–7). See Supplementary Videos 1 and 4.

motion of the tissue–tissue boundary. As classical "healing" assays exclusively focus on collisions between rectangular tissue domains, we first investigated the importance of tissue shape and size in collisions between homotypic tissues (i.e., of the same cell type). Specifically, we characterized interactions between growing pairs of millimeter-scale epithelial tissues, including equal-size rectangles, circles, small vs. large circles, and circles vs. rectangles (Fig. 1 and Supplementary Video 1). We used MDCK epithelial cells, a standard model[5,7], and labeled each tissue in a pair with a different color (Methods) to clearly distinguish the boundary. Imaging over 2–3 days, we observed no mixing (Supplementary Video 2).

Collisions between identical rectangular tissues are well characterized from the traditional wound healing scratch and barrier removal assays[5,33], and our data here confirmed the expected symmetric collision and fusion along the midline (Fig. 1a), allowing us to move on to unstudied configurations. We next compared identical circles, where we observed that a straight boundary formed at the midline as before (Fig. 1b), but it was nearly twice as smooth as the boundary between colliding rectangles (Supplementary Fig. 1, $p$ value 0.014, see Methods). We suspect that this is due to the difference in collision dynamics: while parallel strips of tissue collide all along the collision line at once, circles collide at a single point and gradually extend the boundary line outward (Supplementary Videos 2 and 3).

Finally, we introduced significant asymmetry by replacing one of these circles with either a much larger circle, or a long, thin rectangle (Fig. 1c, d). In each case, we observed a curving of the boundary away from the larger tissue, which was especially

notable in the circle-rectangle collisions. We aligned and averaged the final segmented fluorescence signals that revealed stereotyped and reproducible collision patterns determined solely as a function of initial tissue shapes (Fig. 1, rightmost column).

**Predicting the shape of colliding tissues**. The stereotyped nature of these collision patterns implied that a computational design tool could capture and predict the evolution of tissue shapes upon collisions. We previously established that freely growing epithelia expand outward with a normal velocity $v_n$, which, except in high-curvature zones, is uniform around the perimeter of a tissue and independent of the tissue geometry or density[10]. Here, we incorporated this observation into our model to predict the expansion and interaction of multiple tissues by assuming that tissue edges pin in place upon contact (Supplementary Fig. 2 and Supplementary Note 1). We initialized the model simulations using the initial tissue locations from experiments, and we used $v_n = 29.5\,\mu m/h$ as measured in ref. [10] and confirmed here (Methods). Without any fit parameters, these simulations predict the shape evolution of the colliding tissue pairs in our experiments (Fig. 1 and Supplementary Video 4, blue/yellow/white outlines show model predictions).

Consistent with our observations, pairs of equally sized rectangles or circles produce a straight boundary, while mismatched shapes produce a curved boundary (Fig. 1a–d). In our model, this is because the initial tissue edges are equidistant to the dividing midline in equal tissues but closer to the midline in large tissues than in smaller tissues. In all cases, we found that the mean error of the predicted boundary was compatible with its measured roughness (Supplementary Fig. 1), showing that our modeling approach is appropriate at these scales.

**Homotypic tissue boundary dynamics and collision memory**. Having established a platform for reproducibly studying the formation of complex boundaries between homotypic tissues, we next aimed to connect key biophysical factors to the behavior of these tissue–tissue boundaries. To this end, we studied collisions of homotypic tissues with relative mismatches in either size or cell density. Using the same configuration of two rectangles as in Fig. 1a as a control (Fig. 2a), we prepared tissue pairs with an initial mismatch in tissue width of 1000 vs. 500 μm (Fig. 2b), and other pairs with an initial mismatch in cell density of ~2600 vs. ~1800 cells/mm$^2$ (Fig. 2c).

First, we determined how asymmetry in either tissue width or density affected boundary motion upon collision. We tracked the mean tissue boundary and found that wider and denser tissues displaced narrower and less dense tissues, respectively. Boundary motion was pronounced, directed, and sustained for 15–20 h before stopping (red and blue in Fig. 2d and Supplementary Video 5). In contrast, control experiments with symmetric tissue collisions showed larger boundary fluctuations with very little average drift (black in Fig. 2d and Supplementary Video 5). Prior studies have noted similarly biased boundary dynamics, but only in heterotypic tissue collisions, for example between wild-type and Ras-transformed endothelial cells[31,32]. Here, we show that collisions between homotypic tissues—genetically identical—also produce boundary motion due to asymmetry in tissue size or cell density. In contrast to heterotypic collisions[32]; however, homotypic tissue boundary motion eventually stops.

We related boundary motion to tissue flow using particle image velocimetry (PIV) to measure the velocity field. We represented these data in kymographs of the velocity component along the collision direction, $v_x$, averaged over the tissue length, across multiple tissue collisions (Fig. 2e, see Methods). With identical (control) tissues, cells around the tissue boundary symmetrically

reversed their velocity shortly after collision; convergent motion became divergent. We defined the "center of expansion" as the position from which tissue flow diverges (Methods). For control tissues, the center of expansion lies very near the tissue centroid shortly after collision (Fig. 2e, left).

A key biological question for tissue healing and fusion is at what point, if any, do two fused tissues act as one in terms of their overall dynamics? We investigate this question in collisions between tissues with size or density mismatch, which exhibited tissue flow towards the less dense or narrower tissues. In these cases, the centers of expansion began at the centroid of wider or denser tissues rather than at the overall centroid or collision boundary (Fig. 2e, center and right). The center of expansion then gradually shifted towards the centroid of the fused tissue. After the center of expansion reached the overall centroid, the fused tissue expanded symmetrically without memory of the collision. Thus, by comparing expansion centers to geometric centroids, we demonstrated that a transition exists beyond which two colliding tissues cease functioning independently and shift to behaving as one larger tissue.

**Cell density gradients drive boundary motion**. We hypothesized that collision boundary motion was driven by differences in tissue pressure, which are directly linked to cell density gradients[37–41]. To test this, we quantified local cell density (Methods) and represented it in kymographs for each collision assay (Fig. 2f). In all cases, we found collision boundaries moved down local density gradients, consistent with our hypothesis.

While tissues in the size-mismatch assay had the same initial density, the larger tissue had a higher density at the time of collision (Fig. 2f, center and Supplementary Fig. 3). This observation is consistent with our prior work showing that, even when prepared with the same density, larger tissues develop higher cell densities than smaller tissues as they expand due to increased relative tissue spreading in smaller tissues[10]. Accordingly, initially larger tissues displaced initially smaller tissues. Respectively, in the density mismatch assay, the tissues that are initially denser remain denser at the time of collision (Fig. 2f, right). Accordingly, the density gradient at the onset of collision correlates with the initial density mismatch (Supplementary Fig. 4), culminating in the denser tissues displacing the less dense ones.

To understand the mechanics of the collision boundary motion, we modeled the expanding tissue as an active compressible medium[42]. Tissue expansion is driven by polarized active cell-substrate forces, which are known to be maximal at the tissue edge and decay over a distance $L_c \sim 50\,\mu m$ as we move into the cell monolayer[43,44]. Hence, we ignore active traction forces at the tissue boundary after collision, which is ~1 mm away from the outer tissue edges. At the collision boundary, we establish a force balance whereby pressure gradients drive tissue flow $\boldsymbol{v}$ as

$$-\nabla P = \xi \boldsymbol{v}. \tag{1}$$

Here, $\xi$ is the cell-substrate friction coefficient, which for simplicity we assume to be density-independent. Including a density-dependent friction would affect the speed but not the direction of boundary motion, which is determined by the tissue pressure profile. We assume that the tissue pressure $P$ increases with cell density $\rho$ as specified by an unknown equation of state $P(\rho)$, with $P'(\rho) > 0$. Hence, we obtain

$$\boldsymbol{v} = -\frac{P'(\rho)}{\xi}\nabla\rho, \tag{2}$$

which predicts that the collision boundary moves from high to low cell densities with a speed proportional to the density gradient (Fig. 2g).

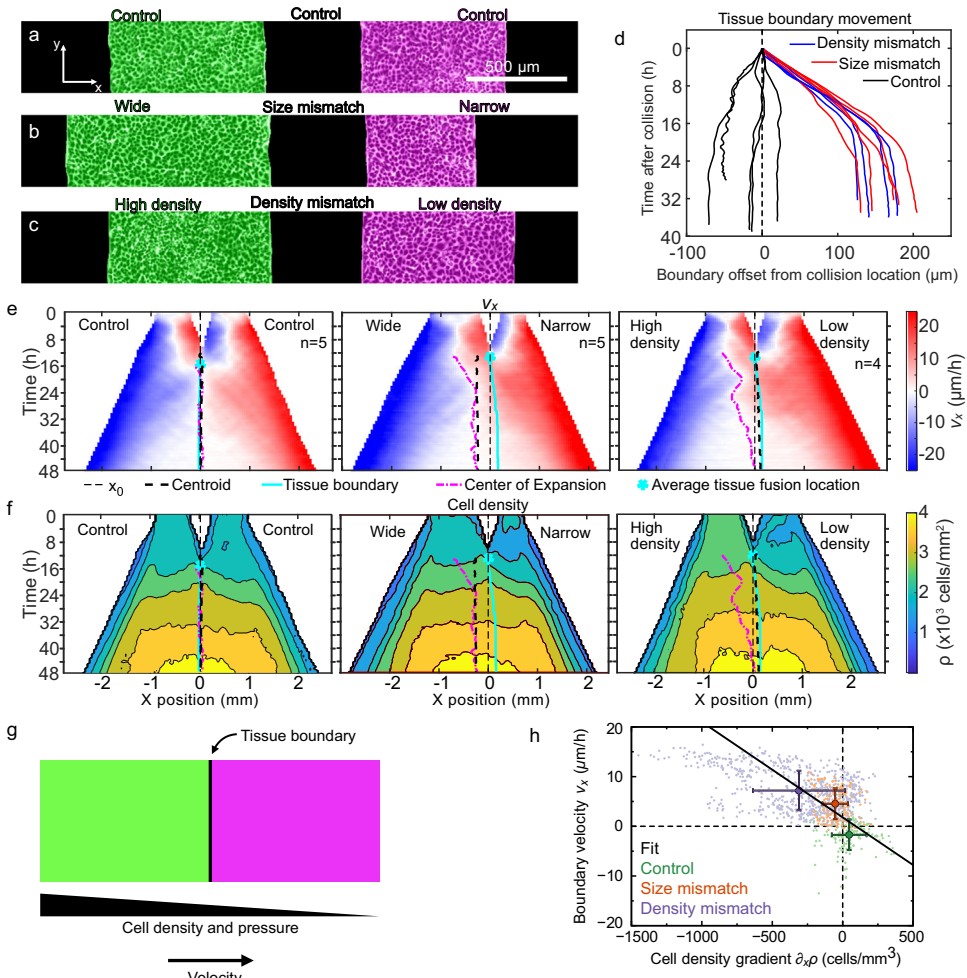

**Fig. 2 Asymmetric tissue collisions produce boundary motion. a–c** Initial condition of collisions between control (**a**), size mismatch (**b**), and density mismatch (**c**) rectangle tissue pairs. Micrographs are representative of $n = 5$ (control), $n = 5$ (size mismatch), and $n = 4$ (density mismatch) independent experiments with similar results. **d** Tissue boundary displacement in both mismatch and control collisions. **e, f** Kymographs of tissue velocity $v_x$ (**e**) and cell density (**f**) along the collision direction for control (left, $n = 5$), size mismatch (center, $n = 5$), and density mismatch (right, $n = 4$) collisions. The superimposed curves indicate initial midline location $x_0$ (thin black dashed line), the geometric tissue centroid (thick black dashed line), the tissue boundary (cyan line), and the center of expansion (pink dash-dotted line), as defined in the text and Methods. The cyan star marks the average tissue fusion location, which occurs at the initial point of the tissue boundary line. Panels **d–f** are from the same data set. **g** Our model proposes that the tissue boundary moves driven by pressure gradients between tissues of different cell densities. **h** Consistent with our model, the velocity and the cell density gradient at the tissue boundary are negatively correlated (correlation coefficient $r = -0.47$). Small points represent individual data, big points correspond to the averages for each of our three assays, and the black line is a linear fit through the averages. This plot includes data from experiments with initial density mismatches higher than in panels **d–f**. The range of applicability of our model extends to all of these density mismatches (additional $n = 16$, Supplementary Figs. 4 and 5). Error bars are standard deviation. See Supplementary Videos 5 and 6.

To test this prediction, we measure both the velocity and the density gradient at the boundary for each experiment in our three different assays (Methods). Consistent with our prediction, the results show a negative correlation between the boundary velocity and the cell density gradient (Fig. 2h). To more strongly connect density gradients to boundary displacement, we explored the importance of both the steepness of the density gradient and the absolute cell numbers for a given gradient. In the first case, we swept the initial density mismatch ratio from 1.3, as in Fig. 2a–f, up to 2.6—the largest mismatch we could experimentally achieve. Higher initial density mismatches led to higher density gradients upon collision, and hence to faster motion of the collision boundary (Supplementary Fig. 5a, b and Supplementary Video 6). Thus, our model applies across all these density mismatches, which are shown together in purple in Fig. 2h. Next, we varied the absolute cell densities of the colliding tissues, ranging from 1200 to 3300 cells/mm$^2$, while preserving the mismatch ratio. We

found that the absolute densities did not significantly affect collision boundary displacement (Supplementary Fig. 5c). Altogether, we conclude that cell density gradients drive boundary motion, which is consistent with pressure-driven tissue flow.

These pressure-driven tissue flows could be driven either by passive volume-exclusion forces between cells, which result from cell crowding, or by active cell-generated forces, which can also be regulated by cell density. To probe the cellular origin of the forces that drive boundary motion, we conducted density mismatch collision assays between tissues treated with blebbistatin (Methods). This treatment inhibits myosin-II activity, hence decreasing active cellular forces[45]. Here, blebbistatin treatment resulted in smaller boundary displacements (Supplementary Fig. 6 and Supplementary Video 7), which suggests that myosin-generated active forces provide a major contribution to boundary motion. Myosin, then, is part of the molecular pathway that translates the density mismatch into a pressure gradient that moves the tissue

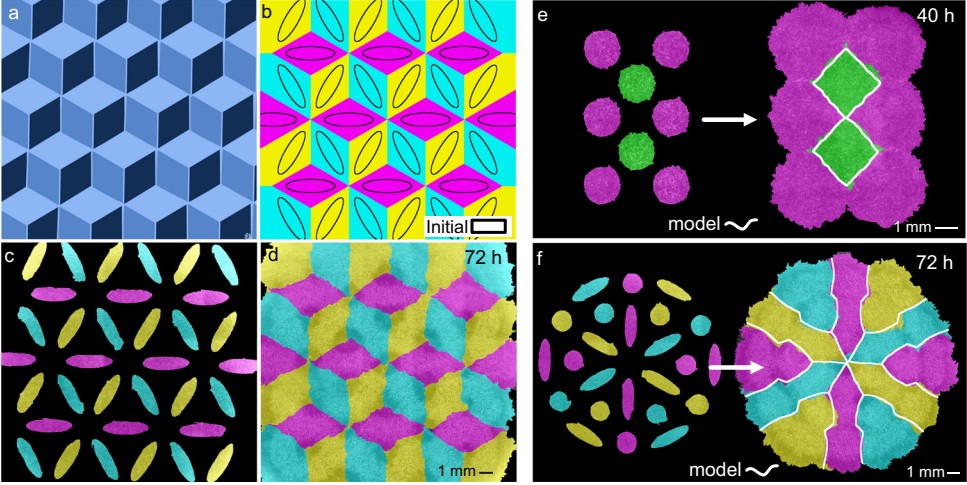

**Fig. 3 TissEllate approach to design complex tissue composites. a**, **b** Target tricolor tessellation (**a**) and chosen initial condition (solid black lines) with accompanying final pattern simulated using *TissEllate* (**b**). **c**, **d** In vitro realization of the tissue composite, which self-assembles from the initial pattern (**c**) to the final tessellation by collision (**d**), representative of three independent experiments with similar results. **e**, **f** Initial pattern and final tessellation for a two-dimensional hexagonal lattice (**e**, one experiment) and another complex pattern (**f**, representative of six independent experiments with similar results). The white outlines indicate the simulated tissue shapes. See Supplementary Videos 8–10.

boundary. Cell density can also regulate cell–cell adhesion strength. Future work can investigate whether the density mismatch induces differences in cell–cell adhesion between colliding tissues, and whether they contribute to the mechanical forces that drive boundary motion. Overall, these results bridge the tissue-scale mechanical picture of our model with some of the molecular mechanisms underlying tissue collision dynamics.

**Estimating tissue mechanical properties from collisions**. Based on our model, we use our measurements of cell density and velocity to extract information about the tissue's equation of state $P(\rho)$. To this end, we obtain the average boundary velocity and density gradient for each assay, and we fit a line to them (Fig. 2h). From this fit, and using $\xi \sim 100\,\text{Pa} \cdot \text{s}/\mu\text{m}^2$ [44,46], we obtain $P'(\rho) \sim 0.5\,\text{Pa} \cdot \text{mm}^2$. This result indicates that, in the conditions of our experiments, for every cell that we add per square millimeter, the tissue pressure goes up about 0.5 Pascal.

Next, we use these results to estimate the mechanical properties of the cell monolayer. To this end, we assume a specific equation of state $P(\rho)$. Whereas the functional form of $P(\rho)$ for tissues remains to be established in experiments, here we use the form proposed in theoretical work by Recho et al.[47]:

$$P(\rho) = K \ln\left(\frac{\rho}{\rho_e}\right), \tag{3}$$

where $K$ is the bulk modulus of the monolayer around the reference cell density $\rho_e$. Accordingly, $K$ is a density-independent parameter that characterizes the linear mechanical response of the cell monolayer around its reference state. Deriving generalized equations of state with a density-dependent bulk modulus $K(\rho)$ remains a challenge for future work. Equation (3) was justified theoretically for growing tissues around their homeostatic state, around which the cell proliferation rate varies linearly with cell density[47]. Although we cannot test the accuracy of this equation of state in our experiments, its assumptions are quite generic and could be met in our system; hence, our choice. From Eq. (3), we have $P'(\rho) = K/\rho$. Using the average cell density measured in our experiments during boundary motion, $\rho = (3.4 \pm 0.2) \times 10^3\,\text{mm}^{-2}$ (SD), we estimate $K \sim 2\,\text{kPa}$.

This order-of-magnitude estimate falls in between two previous measurements of bulk moduli of MDCK cell monolayers. First,

in-plane stretching of suspended cell monolayers yielded a stiffness $E = 20 \pm 2\,\text{kPa}$[48]. Because these monolayers have no substrate, cells do not migrate, and hence suspended monolayers might have different mechanical properties than monolayers on a substrate. Second, in spreading cell monolayers, a linear relationship between tension and strain revealed an effective tensile modulus $\Gamma = 2.4 \pm 0.4\,\text{mN/m}$[49]. Using a monolayer height $h = 5\,\mu\text{m}$[44,50], this value translates into a stiffness $E \approx 0.48\,\text{kPa}$. Complementary to these measurements, which probe tissue stiffness under extension, our estimate reflects the stiffness of the cell monolayer under the compression that results from a tissue collision. The mechanical response of tissues is strongly non-linear. Hence, the response to compression and to extension need not be the same. In fact, the cellular mechanisms responsible for extensional and compressional tissue elasticity can be quite different, and hence the corresponding moduli could differ widely[51]. Here, our estimate suggests that the compressional modulus of MDCK cell monolayers on a substrate is of a similar order of magnitude to the extensional one.

Overall, our collision assays provide a way to probe the bulk mechanical properties of migrating cell monolayers, which are otherwise difficult to measure. Remarkably, analyzing collisions between tissues that differ only in their cell density allowed us to infer mechanical properties without measuring any mechanical forces. Rather, we employed our model to relate tissue flows to pressure and density gradients, from which we inferred the relationship between pressure and density. In the future, collision assays might be used to measure the equation of state of cell monolayers, which is a key input for mechanical models of growing and expanding tissues[42,52].

**Large-scale tissue tessellations for cell sheet engineering**. The stereotyped dynamics of tissue collisions that we uncovered suggested simple underlying design rules that would allow self-assembled tissue tessellations to be designed first in silico and then realized in vitro with potential applications in tissue engineering.

We tested this idea with a tessellation inspired by the artwork of M.C. Escher-a "dice lattice" (Fig. 3a). To design this tessellation, we used the computational model described above (Fig. 1) to simulate many initial tissue array conditions until

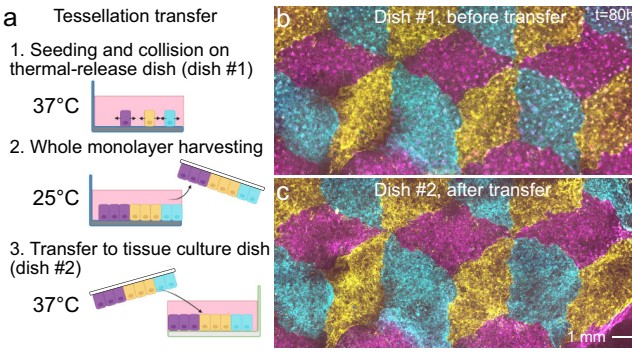

**Fig. 4 Transfer of intact tissue tessellations. a** Schematic of the tissue tessellation transfer process. A support membrane (white) facilitates cell sheet removal and transfer at room temperature from cell culture-ware with temperature-switchable cell adhesion (UpCell). Created with Biorender.com. **b**, **c** Tessellation on UpCell dish (**b**) is transferred to standard tissue-culture dish (**c**). Four independent tessellation transfers were performed with qualitatively similar results.

converging on the pattern of ellipses shown in Fig. 3b. We engineered this pattern with tissues (Fig. 3c) and filmed it developing as predicted (Fig. 3d and Supplementary Video 8). This computer-aided design (CAD) process can be generalized to arbitrary tessellations (Fig. 3e, f and Supplementary Videos 9 and 10), offering a "TissueCAD" approach, which we call "TissEllate", to designing and building composite tissues.

Composite cell sheets may be particularly useful for tissue engineering where cell sheets are extracted from culture vessels and used as building blocks for larger constructs for therapeutic applications[53]. We demonstrated the compatibility of this process with our tissue composites by culturing a dice lattice on a temperature-responsive substrate (UpCell dishes) and then transferring the tissue to a new culture dish (Fig. 4, Methods). The morphology of sharp tissue–tissue interfaces was preserved during the transfer, demonstrating that such tissue composites can, in principle, be handled like standard cell sheets. Interestingly, such multi-tissue composites are similar to current clinical approaches such as suction blister arrays (e.g., the "CelluTome") where 50+ circular epidermal patches from a patient's own healthy skin are transplanted to a burned or injured region and allowed to collectively re-epithelialize and fuse to cover the injury[54–56]. Given the encouraging clinical results of such approaches, the ability to customize internal tissue architecture and coverage we present here may prove valuable in future cell sheet applications.

**Dynamics at tri-tissue collisions**. During the self-assembly of tissue tessellations, we observed a special behavior at tri-tissue collision points: We often found long streams of the inner tissue migrating in between the outer tissues, similar to an extrusion process (Fig. 5a, b). These extruding streams of inner tissue were narrow, progressively necking down to the single-cell scale, visually reminiscent of streams of invasive cancer cells (Fig. 5c–e). However, these events, which we call "escapes", involve the co-migration of all three tissues rather than the invasion of one tissue into the others. In other words, the inner tissue does not break through the outer ones, rather, it squeezes between them as they expand (Fig. 5c–e and Supplementary Videos 11 and 12). Consequently, the initial relative positions of the three colliding tissues are a strong statistical determinant of escape events (Supplementary Figs. 7 and 8).

We characterized the dynamics of escapes by measuring cell speed fields around tri-tissue collisions, which showed that the

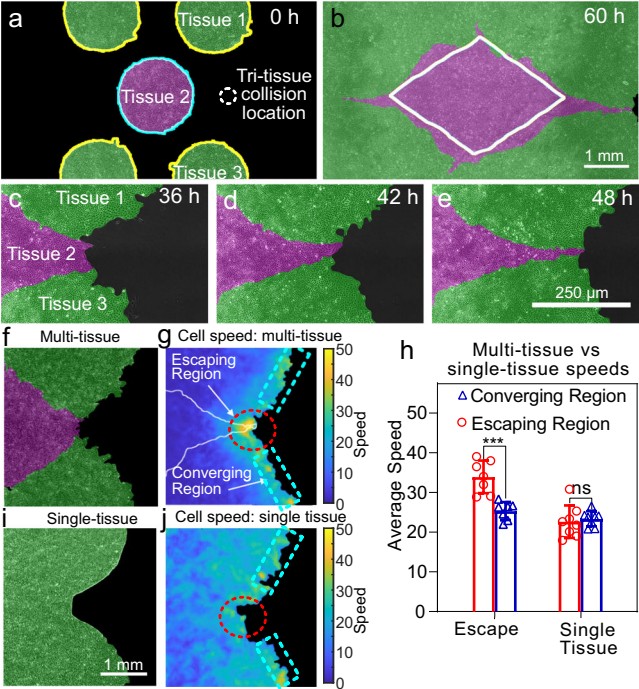

**Fig. 5 Tri-tissue collisions produce "escape" events. a**, **b** Example configuration in which the magenta tissue nearly escapes in between the green tissues during expansion. Labels denote three tissues that participate in a tri-tissue collision leading to an escape event. The white line in **b** indicates the tissue boundary predicted by our simulation tool *TissEllate* based on binary collisions. Escapes occur in 50% of cases, $n = 14$ (Supplementary Fig. 7). **c–e** Close-up view of the escape process, from a different experiment with higher magnification. The same labeling scheme used as in panel **a**. Micrographs show one experiment, as representative example of the 7 of 14 independent experiments that resulted in escapes. **f–h** Tissue dynamics in a tri-tissue collision (**f**) feature a higher local cell speed in the escaping region than the converging region (**g**) with a *p* value of 0.006 using a two-sided Mann–Whitney *U* test (**h**). **i**, **j** A single tissue with a similar pre-escape geometry (**i**) does not have this speed increase in the escaping region (**j**). Plot shows mean with standard deviation across $n = 7$ (escape) and $n = 8$ (single tissue) independent collisions in a single experiment, of which associated micrographs are representative. Source data are provided as a Source Data file. See Supplementary Videos 11 and 12.

escaping inner tissue migrated faster than its neighbors (Fig. 5f–h, see Methods). To determine whether this speed increase was a generic consequence of the local negative curvature of tri-tissue collision points, we compared tri-tissue collisions to a single tissue patterned to match the overall shape of the colliding tissues at the time of escape (Fig. 5i). For the single tissue case, we did not observe any speed increase (Fig. 5h, j), which rules out local curvature as the sole cause of escape events. Therefore, escapes are not a direct result of the overall geometry of the multi-tissue composite; rather, they emerge from the collision interactions between three tissues with their own expansion histories. Furthermore, our simulations of tissue shape evolution based on binary collision rules fail to capture escapes (Fig. 5b). Therefore, escapes are a result of three-body interactions. Moreover, this extrusive behavior represents an unexpected mechanism for producing fine structures between tissues.

**Heterotypic tissue boundary dynamics**. Where homotypic tissues reflect fusion events during morphogenesis and healing, heterotypic tissue boundaries (i.e., different tissue types) arise

during compartmentalization, and our fourth experiment high-lighted how heterotypic tissue interactions can differ from those between homotypic tissues. Critically, heterotypic tissues can have different cell migration speed, motion patterns, and cell–cell interactions (e.g., adhesion and repulsion). However, these differences were not included in our physical model of tissue collisions, and hence we did not use it to analyze heterotypic collisions. Generalizing the biophysical and simulation models to heterotypic collisions is an interesting and important direction for future work.

Here, we prepared co-cultures of the breast cancer cell lines MCF10A (benign), MCF7 (malignant, non-invasive), and MDA-MB-231 (metastatic) as monolayers of the same size and cell density. We used homotypic MCF10A collisions as a reference, for which we observed non-mixing and boundary dynamics similar to the homotypic MDCK collisions discussed earlier (Supplementary Video 13).

We first collided rectangular monolayers of MCF10A and MDA-MB-231 cells, which have the largest phenotypic difference among the cell lines we used. While these tissues have similar expansion speeds, they exhibit radically different collective dynamics, reflective of different cell–cell adhesion strengths[57] (Supplementary Video 14). While cells in MCF10A tissues hardly exchange neighbors, the metastatic MDA-MB-231 cells continually undergo neighbor exchanges and even crawl over each other out of the plane. Upon collision, the MCF10A tissue simultaneously displaced and crawled underneath the MDA-MB-231 tissue (Fig. 6a–d).

We next investigated collisions between MCF10A and MCF7 monolayers. The MCF7 monolayer expands about 6 times more

slowly than the MCF10 monolayer, which allowed us to explore the effects of different edge speeds on tissue collisions, with implications for multi-tissue design. Surprisingly, we found that the slower MCF7 tissues actually displaced the MCF10A tissues (Fig. 6e and Supplementary Video 15), which may be due to differences in cell–cell and cell-substrate adhesion. Thus, a higher expansion speed does not imply the ability to displace tissues that expand more slowly. In fact, MCF10A cells at the collision boundary reversed their velocity and migrated away from the MCF7 tissue within 8 h after collision, starting at the boundary and progressively moving into the MCF10A monolayer (Fig. 6f). This behavior seems a tissue-scale analog of the cellular behavior known as contact inhibition of locomotion, whereby a cell stops and reverses its direction of motion upon collision with another cell[2–4].

Furthermore, in collisions between tissues with different expansion speed, the faster tissue should be able to engulf the slower tissue, similar to the engulfment between tissues with differential adhesion[58]. We confirmed this hypothesis in collisions between rectangular strips of MCF10A cells (fast) and circles of MCF7 cells (slow), which we reproduced with our tissue shape model by incorporating different speeds into our simulation (Fig. 6g–j and Supplementary Video 16, see Supplementary Fig. 9 and Supplementary Note 2). Future work is needed to elucidate the biophysical properties of heterotypic tissue interfaces, but here we highlight how differences in expansion speed enable design options in future multi-tissue studies and applications.

## Discussion

We investigated how tissue–tissue interactions can be harnessed to self-assemble complex composite tissue sheets—tissue tessellations. First, we demonstrated that colliding tissues change shape in stereotyped and predictable ways. Then, we proposed a physical model of tissue–tissue collisions where the critical driver of boundary dynamics is the underlying gradient in cell density across the interface that induces pressure-driven tissue flow down the gradient. Furthermore, we used this model to estimate the material properties of the colliding tissues without any force measurements. In the future, this approach might be used to reveal information about the equation of the state of living tissues.

Regarding the collision dynamics, previous work had shown that heterotypic tissues can displace each other upon collision[31,32]. Our findings revealed that even homotypic tissues, which are genetically identical, can displace each other based on purely mechanical differences, suggesting a mechanical tug-of-war where cellular pressure gradients across the boundary can compete with tissue-scale contact-inhibition-of-locomotion. Therefore, our collision assays could be used to study mechanical tissue competition[37–41,59,60], which might provide biophysical insight into development[61,62], homeostasis[63], and tumor growth[37,59]. In our in vitro experiments, the mechanical competition arises from differences in cell density between the colliding tissues, which are imposed in our experimental setup. In vivo, cell density gradients could potentially arise from a number of biological mechanisms. Uncovering such mechanisms and their consequences for tissue collisions in vivo is an interesting avenue for future research.

Furthermore, based on the reproducible and almost algorithmic tissue interactions that we found, we developed computational design tools to create complex tissue tessellations. Whereas previous work had exclusively studied binary head-on collisions, the tessellations allowed us to study collisions between myriad, arbitrary tissues. For example, a finding unique to tri-tissue interactions is that the inner tissue can speed up and

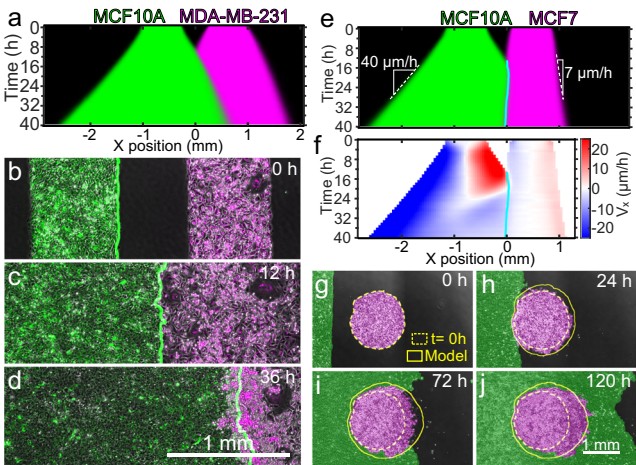

**Fig. 6 Heterotypic tissue collisions. a** Average kymograph of segmented fluorescence for collisions ($n = 12$) between rectangular MCF10A and MDA-MB-231 tissues of the same size and cell density. The MCF10A tissue displaces the MDA-MB-231 tissue. **b–d** Snapshots of the co-culture at the initial configuration (**b**), the collision time $t = 12$ h (**c**), and 24 h after collision, i.e., $t = 36$ h (**d**). The green line indicates the edge of MCF10A tissue. Micrographs are representative of 11 independent tissue collisions with similar results. **e, f** Average kymographs of segmented fluorescence (**e**) and velocity $v_x$ (**f**) for collisions ($n = 11$) between rectangular MCF10A and MCF7 monolayers of the same size and cell density. The cyan line indicates the tissue boundary. **g–j** Snapshots of the initial configuration (**g**), collision onset (**h**), partial engulfment (**i**), and full engulfment (**j**) of a circular MCF7 (magenta) tissue by a rectangular MCF10A (green) tissue. Simulations (yellow outlines) confirm that a difference in expansion speed is sufficient to predict the engulfment process. Engulfment repeated with similar results in two independent experiments. See Supplementary Videos 13–16.

squeeze between the outer tissues. Such collision dynamics do not follow binary collision rules and are particularly surprising in their ability to produce complex, fine patterns almost at the single-cell scale. Our work thus opens the door to characterizing and understanding the dynamics of increasingly complex multi-tissue collisions.

Finally, the tessellations are obtained by self-assembly, which allows the tissue boundaries to develop naturally. Thus, our work demonstrates how tissue sheets can be treated as "designable" living materials. Specifically, we developed a simulator that, despite lacking both biophysical laws and cellular resolution, predicts tissue patterns accurately at the 100+ µm scale. This feature makes the simulator useful to design tissue composites in silico before realizing them in vitro. This approach is compatible with advanced biofabrication strategies such as cell sheet engineering, which we demonstrated by transferring an "Escher" tissue sheet between Petri dishes while preserving tissue integrity and internal boundaries. Tissue tessellation should also be compatible with bioprinting, which could be used to pattern larger arrays of the initial tissue seeds.

While our work focused on the biophysical elements of tissue interactions, our tissue arraying method provides unique advantages relative to other forms of tissue patterning. First, razor-writing allows custom silicone stencils to be cut and seeded within minutes, all at the benchtop. Second, stencils fundamentally allow cells to grow out simply by removing the stencil, which is distinct from many advanced tissue patterning processes. For instance, printing of unique DNA-oligomer pairs or patterned extracellular matrix allows for precise, high throughput patterning of micro-tissues with unique composition, but does not support outgrowth[64–66]. Similarly, elegant photopatterning methods allow for precise, spatiotemporal control of the microenvironment but require significant infrastructure and are difficult to scale up to many tissues[67,68]. Ultimately, while razor-writing has limitations below 500 µm, its convenience and low cost make it extremely accessible to the numerous and wide range of research groups studying monolayer biophysics and cell sheet engineering.

Finally, we envision future work generalizing these design tools to tissue interactions with heterotypic tissues and 2D vs. 3D geometries, each of which will necessitate characterizing boundary rules and implementing them in the computational tools. There is also merit in comparing the similarities and differences between interface formation between growing tissues and immiscible, viscous fluids where similar tessellation dynamics can arise[69]. Such extensions will allow the design of increasingly complex, multi-functional tissue patches for biophysical and biomedical applications.

## Methods

**Cell culture**. MDCK-2 wild-type canine kidney epithelial cells (ATCC) were cultured in customized media consisting of low-glucose (1 g/L) DMEM with phenol red (Gibco, USA), 1 g/L sodium bicarbonate, 1% streptomycin/penicillin (Gibco, USA), and 10% fetal bovine serum (Atlanta Biological, USA). MCF10A human mammary epithelial cells (ATCC) were cultured in 1:1 DMEM/F-12 (Thermo Fisher Scientific, USA) media that consists of 2.50 mM L-Glutamine and 15 mM HEPES buffer. This media was supplemented with 5% horse serum (Gibco, New Zealand origin), 20 ng/mL human EGF (Sigma, USA), 0.5 µg/mL hydrocortisone (Fisher Scientific), 100 ng/mL cholera toxin (Sigma), 10 µg/mL insulin (Sigma, USA), and 1% penicillin/streptomycin (Gibco, USA). MDA-MB-231 (ATCC) and MCF7 (ATCC) human mammary cancer cells were both cultured in 1:1 DMEM/F-12 (Thermo Fisher Scientific, USA) media supplemented with 10% fetal bovine serum (Atlanta Biological, USA) and 1% penicillin/streptomycin (Gibco, USA). All cells were maintained at 37 °C and 5% $CO_2$ in humidified air.

**Tissue patterning and labeling**. Experiments were performed on tissue-culture plastic dishes (BD Falcon, USA) coated with type-IV collagen (MilliporeSigma, USA). Dishes were coated by incubating 120 µL of 50 µg/mL collagen on the dish under a glass coverslip for 30 min at 37 °C, washing three times with deionized distilled water (DI), and allowing the dish to air-dry. Stencils were cut from 250-

µm-thick silicone (Bisco HT-6240, Stockwell Elastomers) using a Silhouette Cameo vinyl cutter (Silhouette, USA) and transferred to the collagen-coated surface of the dishes. Cells were separately labeled using CellBrite™ (Biotium, USA) red and orange dyes for two-color experiments and CellBrite™ (Biotium, USA) red, orange, and green dyes for three-color experiments.

For MDCK experiments, suspended cells were concentrated at ~2.25 × 10^6 cells/mL and separated into identical aliquots for labeling. We added 8 µL of the appropriate membrane dye color per 1 mL of media and briefly vortexed each cell suspension. Then, we immediately pipetted into the stencils at ~1000 cells/mm², taking care not to disturb the collagen coating with the pipette tip. To allow attachment of cells to the collagen matrix and labeling of the cell membranes, we incubated the cells in the stencils for 30 min in a humidified chamber before washing out the dye and filling the dish with media.

For experiments using other cell types, suspended cells were concentrated at ~3 × 10^6 cells/mL and 10 µL of membrane dye color per 1 mL of media. We briefly vortexed the cell suspension and allowed it to incubate for 20 min at 37 °C. We then centrifuged the suspension and removed the supernatant, replacing it with the appropriate media without dye. We pipetted into the stencils at the same volume as before (greater number of cells), and incubated the cells in the stencils for 2–3 h to allow attachment before filling the dish with media.

For all experiments, we then incubated the cells for an additional 18 h after cell attachment to form confluent monolayers in the stencils. Stencils were removed with tweezers, with imaging beginning ~30 min thereafter. Media without phenol red was used throughout seeding and imaging for three-color experiments to reduce background signal during fluorescence imaging. For experiments involving blebbistatin (BioGems, USA), 25 µm was mixed into fresh media and added to the dish 2 h after the stencils were removed. To ensure no phototoxicity, no excitation wavelengths below 500 nm were used during live imaging[70]. This was confirmed in our assays by ensuring migration speeds of blebbistatin-treated tissues matched that of control tissues.

**Live-cell time-lapse imaging**. We performed imaging on an automated, inverted Nikon Ti2 with a Nikon Qi2 CMOS camera and NIS Elements software (version AR 5.20.01 64-bit). We equipped the microscope with environmental control for imaging at 37 °C and humidified 5% $CO_2$. Final images were composited in NIS Elements from montages of each pair or tessellation.

For experiments from Fig. 5c–j, we used a 10× phase-contrast objective to capture phase-contrast and fluorescence images every 10 min. RFP/Cy5 images were captured at 10% lamp power (Sola SE, Lumencor, USA) and 150 ms exposure time. No phototoxicity was observed under these conditions for up to 24 h.

For all other experiments, we used a 4× phase-contrast objective to capture phase-contrast images every 20 min. For two-color time-lapse images, RFP/Cy5 images were also captured every 20 min at 15% lamp power (Sola SE, Lumencor, USA) and 500 ms exposure time. For three-color time-lapse images, RFP/Cy5 images were captured every 60 min at 15% lamp power (Sola SE, Lumencor, USA) and 300 ms exposure time, while GFP images were captured every 120 min at 15% lamp power (Sola SE, Lumencor, USA) and 300 ms exposure time. No phototoxicity was observed under these conditions for up to 72 h. Final images were composited in NIS Elements from montages of each pair or tessellation.

**Tissue dye segmentation**. The tissue dye becomes diluted as cells divide and spread, so the dye at tissue edges (where cells are more spread and divide more frequently) becomes much more dim than the center of tissues. Because we saw no mixing in our collisions, we segmented the fluorescence channels using a custom MATLAB (Mathworks, version 2018b) script and overlaid them with the phase-contrast images for clear visualization. To segment fluorescence images, we normalized the fluorescence channel histograms to each other and compared relative brightness for each pixel between channels. We then masked with the binary masks obtained from the phase-contrast channel.

**Setting $v_n$ for model**. We set the normal velocity for the model (all shapes and tessellations) according to the outward velocity of the outer edges of the control rectangle collisions. The outward velocity was found to be 29.4 ± 2.3 µm/h (standard deviation), so we used the previously reported speed for expanding circles of 29.5 µm/h[10].

**Velocity measurements**. We calculated tissue velocity vector fields from phase-contrast image sequences, rotating each image so that the initial tissue locations in image pairs were horizontal. We used the free MATLAB package PIVLab version 2.56 with the FFT window deformation algorithm, employing a first pass window size of 96 × 96 pixels and second pass of 48 × 48 pixels, with 50% pixel overlaps. This resulted in a final window size of 88 × 88 µm. Data were smoothed in time with a moving average of three timepoints centered at each timepoint.

**Average kymographs**. We first constructed kymographs of each rectangular collision pair, averaging over the vertical direction of each timepoint and ignoring the top and bottom 1 mm. We then averaged the individual tissue kymographs, aligning by the initial tissue configuration, and determined the edge extent from the median extent of the individual kymographs.

**Center of expansion**. We determined the center of expansion by thresholding as $|v_x| < 3$ μm/h for individual kymographs of $v_x$. We filtered for the largest contiguous region and took the midline of this region as the center of expansion.

**Cell density measurements**. We first reproduced nuclei positions from 4× phase-contrast images using our in-house Fluorescence Reconstruction Microscopy tool[71]. The output of this neural network was then segmented in ImageJ to determine nuclei footprints and centroids. Local density was calculated for each PIV window by counting the number of nucleus centroids in that window.

**Boundary velocity and cell density gradient determination**. Boundary velocity was found from the position change of the midline in Fig. 2d. Cell density gradient $\partial_x\rho$ was found as $\frac{\rho_R - \rho_L}{x_R - x_L}$, where $\rho_L$ and $\rho_R$ are the total density within 300-μm-wide regions immediately to the left and right of the tissue boundary, respectively, and $x_R - x_L = 300$ μm. We plotted $\partial_x\rho$ for timepoints between 20 and 36 h, which is after collisions and before the boundary stops moving.

**Statistical analysis**. For all significance testing and corresponding $p$ values reported in this study, we used GraphPad Prism 9.1.0 (GraphPad Software) to perform non-parametric, unpaired two-sided Mann–Whitney $U$ tests. All measurements were taken from distinct samples, where averages and standard deviations were calculated across the series of replicates. In every assay, sample sizes include at least four replicates, across at least three independently performed experiments.

**Cell sheet engineering tissue patterning and transfer**. We first patterned tissues on a 3.5-cm NUNC™ UpCell™ dish with a supportive membrane (Thermo Fisher Scientific, USA). We followed the same collagen coating and stencil application process as before, but passivated the underside of our stencils to avoid damaging the UpCell™ surface. To passivate the stencils, we incubated them for 30 min at 37 °C in Pluronic™ F-127 solution (Thermo Fisher Scientific, USA) diluted in PBS to 2%. We washed the stencils three times in DI and gently dried them with compressed air before transferring them to the dish.

After the tissues reached confluence within the stencils, we removed the stencils and allowed the tessellation to collide and heal for ~72 h. To release the tessellation monolayer from the dish, we changed to cold media and moved the dish to an incubator set to 25 °C for 1.5 h. We then pre-soaked the supportive membrane in the media to avoid membrane folding, and floated the membranes on the media above the tessellations. We then carefully aspirated the media from beside the membranes to ensure tight contact between the membrane and monolayer with no bubbles. We moved the UpCell™ dish with membrane to a 4 °C refrigerator to ensure total release, and prepared a standard 3.5 mm tissue culture dish (BD Falcon, USA) coated with collagen IV as before and filled with warm media. After 7 min at 4 °C, we carefully removed the membrane and tessellation monolayer from the UpCell™ dish and floated it in the tissue culture dish with the tessellation side down. We aspirated the media from beside the membrane to initiate bubble-free contact with the dish surface and covered the membrane with 350 μL of warm media. We incubated the membranes at 37 °C overnight before floating the membrane off the surface by filling the dish with media and removing it with tweezers.

**Reporting summary**. Further information on research design is available in the Nature Research Reporting Summary linked to this article.

## Data availability
The complete raw data generated during the current study are available from the corresponding authors upon reasonable request due to the size of the datasets (terabytes). Representative data and analysis code can be found at https://github.com/CohenLabPrinceton/TissEllation. Source data are provided with this paper.

## Code availability
Key code used in this study is available on our laboratory repository [https://github.com/CohenLabPrinceton/TissEllation.git].

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

## Acknowledgements
This work was supported in part by the National Institutes of Health 1R35 GM133574-01 (to D.J.C.). R.A. acknowledges support from the Human Frontier Science Program (LT000475/2018-C). The authors thank Jenna Heinrich for artwork.

## Author contributions
M.A.H., A.E.W., and D.J.C. designed and performed experiments. M.A.H., R.A., A.E.W., and D.J.C. analyzed data. M.A.H., R.A., and A.K. developed the models. D.J.C. and A.K. funded and supervised the work. M.A.H., R.A., A.E.W., D.J.C., and A.K. wrote the paper.

## Competing interests
The authors declare no competing interests.
