## [Peer Review File · Nature Communications]

REVIEWER COMMENTS

Reviewer #1 (Remarks to the Author):

The manuscript by Heinrich et al. examined the dynamics of cell monolayer and in particular, collisions between cell monolayers with different geometries, cell densities and cell types. The modeling of this behavior mimics experimental observations and the authors suggest that monolayer elasticity may be extracted (though the values are not experimentally verified in this manuscript). As a part of providing potential applications, the authors design and create various geometric patterns and demonstrate these cell monolayers can be harvested and transferred to a new dish. Finally, the escaping of cell layer sandwiched between neighboring monolayers are examined (a behavior not predicted by the model). Every figure in this manuscript is thought provoking and has potentially transformative applications, unfortunately however, not well developed and molecular details are missing. For example, heterotypic tissue collisions using MCF10A vs MDA/MCF7 cell lines demonstrate two divergent behaviors. The authors suggest the displacement of MCF10A cells by MCF7 cells may be due to cell-cell adhesion and eph/Ephrin signaling, but no experimental data to support this hypothesis. I wish the authors have provided more molecular details/explanations for the observed collision events. Perhaps, a broad presentation style of the current manuscript may appeal to the wide-range of audience of this journal. As a cell biology researcher, I felt the lack of molecular details/explanations was somewhat disappointing.

Minor comments: When grown at a high cell density, MDCK cell monolayer will eventually enter highly compacted quiescent state, in some cases, will dorm. I am not sure they will fully polarize with microvilli as implied in the cartoon in Figure 5 on the coverslips, unless they are grown on a filter substrate. In heterotypic tissue collisions, are there possible chemical or physical interactions between different cell types (via soluble factor or cadherins) that dictate the collision behavior? What are the key parameters that define the “strength” of collision (line 278)?

Reviewer #2 (Remarks to the Author):

This study examined the responses following the collision of neighboring epithelial cell colonies as they expand. Using colonies of different sizes, shapes, densities, and cell types, it was concluded that the response for homotypic colonies was determined solely by cell density gradient. Mechanical modeling was then applied to estimate the elastic modulus of the colony, based on the speed of border movement as a function of density gradient. For heterotypic colonies, it was necessary to invoke a

different mechanism driven by differential cell-cell adhesions. The authors then proposed that complex tissues may be engineered by tessellation of cell colonies.

Overall, the experimental ideas are refreshing. However, the strong commitment to a density driven mechanism is premature and potentially misleading. In addition, the mechanical model involves hidden risky assumptions. Overall, the study attempts to cover multiple scattered areas without any of them being sufficiently mature or convincing. Extensive revision with possibly substantial changes in thinking is recommended.

Specific comments:

1. A major concern is the over-commitment to the model that the event is "solely based on cell density gradient", without considering the possibility that cell density gradient may be secondary as a downstream result of a primary mechanism. When addressing heterotypic dynamics where cell density failed to explain the observations, the authors then had to invoke cell-cell adhesion as the driving mechanism. There were also difficulties in explaining the results with colonies of mismatched sizes based on cell density, where the argument that the smaller colony somehow had a low cell density also appeared unconvincing.

A more plausible and unified hypothesis would treat cell-cell adhesion as the primary parameter, which may affect mechanical properties of the colony to determine the outcome for both homo- and heterotypic collisions; small, sparse colonies or weak cell-cell adhesions may lead to weaker integrated mechanical strength, against larger, denser colonies or strong cell-cell adhesions. The importance of cell-cell adhesion in tissue organization was in fact addressed decades ago in the elegant studies of Malcolm Steinberg (1930-2012, at Princeton!), which the authors should carefully consider.

2. The mechanical model as expressed in equation 1-3 has underlying, risky assumptions that must be spelled out and justified. The most concerning is that elastic modulus of a colony is a constant for a given cell type, while in fact it is likely a function of both cell density and cell-cell adhesion; weak or sparse cell-cell adhesion would likely lead to a low modulus and/or viscous behavior. Thus, the derivative of K with respect to ρ should not be treated as a constant in the equation for P' . This caveat casts doubt on the validity of the estimation of elastic modulus.

3. It was proposed that monolayer tessellation can serve as an engineering tool. However, the application is premature due to the poor understanding of heterotypic collision, which is required for the construction of complex tissues. While the proposed process works for homotypic monolayers, the construct is no different from what can be obtained more easily by placing cells on micropatterned substrates.

4. It is unclear what exactly was learned from tri-tissue collisions. The "escape" may be easily explained when an expanding colony (purple cells in Fig. 6) encounters a barrier with a region of weak adhesion (where two green colonies meet in Fig. 6). Also, as a minor comment, the term and the figure are confusing in the sense that panel 6a shows 5 colonies with 2 colors, where the meaning of "tri"-tissue needs to be figured out between the lines.

5. (minor) Figure 2, it would be helpful to mark the points of colony fusion in panels e and f. Clarify if e and f are from the same set of data. Indicate the center of expansion in f.

In summary, the study starts with novel and very promising experiments, but the subsequent sections scattered into various territories with significant caveats. In addition, the role of cell-cell adhesion must be carefully considered for both homo- and heterotypic collisions, which may lead to a unified model. In addition, the sections on tessellation and cell escape, which serves as no more than shiny objects at this stage, should be removed.

Reviewer #3 (Remarks to the Author):

Heinrich and co-workers studied the dynamics of collision of multicellular tissues. They combined experiments and modeling to predict how the interface between two colliding tissues depends on tissue shape and cell density. They simulated the expected position of the collision front for various combinations of rectangles and circles using a previously published method and showed that the predicted front fit well the experimental findings. The main result is that the collision front does not move once the cells collide. By looking at genetically identical tissues that varied in size and density, they showed that density gradient is the main control parameter over the motion of the collision front. They modeled the tissue as a dry active compressible elastic material. They recovered that the collision boundary moves from high to low densities with a speed that is proportional to the density gradient.

Next, they studied heterotypic tissue collision. They used three breast cancer cell lines: MCF10A (benign), MCF7 (malignant, non invasive) and MDA-MB-231 (metastatic) cells. They showed that higher expansion speed does not imply higher collision strength.

Finally, the authors develop a new design tool named Tissellate, which allows them to design complex tissue tessellation based on their initial model for tissue expansion. They showed that different initial seeding geometry leads to predictive self-assembled tissue tessellations. They showed that tri-tissue collision can lead to escape events, where cells within the escape region can move faster than any other cells in the expanding front.

I think this is an elegant paper that presents novel and robust results. The results and the software developed would be of interest for physicists, biologists and bioengineers interested in tissue mechanics and collective cell migration. The manuscript was clear and well organized. I just have a few comments and suggestions below that might strengthen the paper, in particular the comparison between experiments and theory.

Major comments:

1. Justification of the model assumptions could be improved.

Why is the friction independent of density?

Experimental justification for choosing the specific equation of state beside theoretical work by Recho et al.

Why doesn't K depend on cell density?

How does the density at the collision front depend on the initial seeding densities?

2. The comparison between the model and the experiments could be strengthened.

At the moment, the authors only show that velocity and cell density gradient are negatively correlated. While they derive an equation of state for the tissue, I have not seen a direct comparison between the experimental plots (Fig 2H) and the equation of state (equations 2 and 3). More quantitative comparison would strengthen the claim that mechanical properties of a mono layer can be infer by studying the kinematic of the colliding fronts.

Suggested experiments to strengthen the comparison between experiments and theory:

A) increase the density mismatch between the two colliding tissues (right now it is only 1.5X) this parameter could be explored experimentally.

B) Does the absolute density also impact the mechanical properties of the tissue? To test that, maybe the authors could explore the collisions of two tissues with the same density gradients as before, but with a different cell densities on both sides.

Could the authors comment on why they think there is a 2 orders of magnitude difference between the stiffness under compression and the stiffness under extension?

3. The connection between the results from the homotypic collision experiments + model with the heterotypic collision experiments is a bit weak.

This section is disconnected from the others. Can the model derived for MDCK cells help identify what physical properties of the homotypic tissues drives the motility of the colliding front?

Minor comments:

1. The caption for figure 3D says the picture was taken 24h after removing the stencil while the text in the figure says 36h.

2. There is no mention of what cell types are/can be modeled with the Tissellate software. I assumed MDCK cells with identical seeding densities. Can this be extended to other cell types? To heterotypic collision?

Response to reviewers: Heinrich et al. 2021

We appreciated the feedback from the reviewers, and the resulting resubmission adds significant improvements based on this feedback. Our responses, in blue, comprehensively address each Reviewer comment.

Reviewer #1 (Remarks to the Author):

The manuscript by Heinrich et al. examined the dynamics of cell monolayer and in particular, collisions between cell monolayers with different geometries, cell densities and cell types. The modeling of this behavior mimics experimental observations and the authors suggest that monolayer elasticity may be extracted (though the values are not experimentally verified in this manuscript). As a part of providing potential applications, the authors design and create various geometric patterns and demonstrate these cell monolayers can be harvested and transferred to a new dish. Finally, the escaping of cell layer sandwiched between neighboring monolayers are examined (a behavior not predicted by the model).

Every figure in this manuscript is thought provoking and has potentially transformative applications, unfortunately however, not well developed and molecular details are missing. For example, heterotypic tissue collisions using MCF10A vs MDA/MCF7 cell lines demonstrate two divergent behaviors. The authors suggest the displacement of MCF10A cells by MCF7 cells may be due to cell-cell adhesion and eph/Ephrin signaling, but no experimental data to support this hypothesis. I wish the authors have provided more molecular details/explanations for the observed collision events. Perhaps, a broad presentation style of the current manuscript may appeal to the wide-range of audience of this journal. As a cell biology researcher, I felt the lack of molecular details/explanations was somewhat disappointing.

1. We appreciate that the reviewer found this work thought provoking, and we agree that there is significant cell biology to explore here. Our focus was not the underlying molecular mechanisms but rather the biophysical principles and engineering applications of tissue collisions. Yet, to provide an initial exploration of the molecular mechanisms underlying the collision dynamics, we have performed additional experiments treating colliding tissues with blebbistatin to probe the role of myosin-II in driving the motion of the collision boundary (see lines 250-263 in the text). **We found that treating density-mismatched tissues with blebbistatin resulted in smaller displacements of the collision boundary (Fig. S6), which indicates that myosin-generated active forces contribute to boundary motion. Myosin, then, is part of the molecular pathway that translates the density mismatch into a pressure gradient that moves the tissue boundary. This result bridges the tissue-scale mechanical picture of our model with some of the molecular mechanisms underlying tissue**

collision dynamics. Therefore, we believe that it makes our paper more appealing to the broad community at the intersection of biology, biophysics, bioengineering, and active matter. While there are certainly deeper cell biology questions to follow-up with, they are beyond the scope of this manuscript and we defer them to future work.

2. Heterotypic collisions are not expected to follow the same dynamics as homotypic tissues, which effectively form single tissues after colliding. While tissue-scale contractile forces are important in homotypic collisions (as discussed above), these forces will not cross a non-adhesive heterotypic tissue boundary. Different tissue types will also have different homeostatic densities, bulk moduli, and friction coefficients, which are the parameters of our physical model. Moreover, different tissue types can also exhibit differences in cell-cell adhesion and traction forces, as shown, for example, in Refs. 31-33. All these ingredients are not included in our physical model, and therefore we did not use it to understand heterotypic collisions. Rather, we used heterotypic collisions to demonstrate additional potential tessellation design options for tissues with different properties. To make this clear, we moved the section on heterotypic collisions at the end, below the section on tissue tessellations, and added additional language that clarifies the scope of heterotypic collisions (see lines 399-413). While the molecular details governing heterotypic collisions are of interest, they lie beyond the scope of this paper. Hence, **we have taken out suggestions for the molecular signaling basis of heterotypic collision dynamics.**

“Minor comments: When grown at a high cell density, MDCK cell monolayer will eventually enter highly compacted quiescent state, in some cases, will dorm. I am not sure they will fully polarize with microvilli as implied in the cartoon in Figure 5 on the coverslips, unless they are grown on a filter substrate.”

Response: This is a good point to raise. It may be true in other epithelia, but with MDCKs the markers for microvilli such as podocalyxin still localize to the apical surface even on glass substrates. See Cohen, Glocrich, and Nelson PNAS 2016 (Ref. 22). However, we did not image the tissue to assess microvilli while the tissues were on the transfer coverslip membranes, so we removed the villi from the apical surface of the cartoon monolayers in what is now Fig. 4a to avoid any implicit claims.

“In heterotypic tissue collisions, are there possible chemical or physical interactions between different cell types (via soluble factor or cadherins) that dictate the collision behavior?”

Response: While the molecular and physical details governing *heterotypic* collisions are outside the scope of this paper, we will comment briefly here. Both chemical and

physical interactions would be expected to influence boundary motion in heterotypic collisions. Regarding chemical interactions, different cell types exhibit various gene expression profiles and signaling cascades, leading to diverging phenotypes. For one example, see Ref. 27, where cancer cells failed to undergo contact inhibition of locomotion upon contact with non-cancer cells. This was found to be due to eph-receptor-mediated signaling, and may explain why the MCF7 cells in our experiments did not significantly change their migration after collision, while the MCF10A cells *reversed* their velocity in their heterotypic collision (Fig. 6e-f in the revised manuscript). Regarding physical interactions, different cell types can exert different traction forces and can have different cell densities, tissue stiffness, and cell polarization dynamics. All these differences can influence border motion upon collision. For example, Ref. 32 reported that a difference in traction forces drives border displacement upon collision.

“What are the key parameters that define the “strength” of collision (line 278)?”

Response: By a higher “strength”, we meant that a “stronger” tissue is one that could displace another upon collision. To avoid confusion, we rephrased the sentence to read “This shows that a higher expansion speed does not imply the ability to displace tissues that expand more slowly” (line 441).

2. Reviewer #2 (Remarks to the Author):

This study examined the responses following the collision of neighboring epithelial cell colonies as they expand. Using colonies of different sizes, shapes, densities, and cell types, it was concluded that the response for homotypic colonies was determined solely by cell density gradient. Mechanical modeling was then applied to estimate the elastic modulus of the colony, based on the speed of border movement as a function of density gradient. For heterotypic colonies, it was necessary to invoke a different mechanism driven by differential cell-cell adhesions. The authors then proposed that complex tissues may be engineered by tessellation of cell colonies.

Overall, the experimental ideas are refreshing. However, the strong commitment to a density driven mechanism is premature and potentially misleading. In addition, the mechanical model involves hidden risky assumptions. Overall, the study attempts to cover multiple scattered areas without any of them being sufficiently mature or convincing. Extensive revision with possibly substantial changes in thinking is recommended.

Response: We appreciate that the reviewer found our experimental ideas refreshing. We have revised the manuscript extensively, including new experiments, rewriting, and changes in the order and structure to address the reviewer's concerns.

Specifically, we have clarified that our current model applies only to homotypic collisions, which will avoid misleading interpretations as mentioned by the reviewer. To this end, we have moved the section on heterotypic collisions to the end of the manuscript, after discussing tessellations. Together with the revised text, this helps to clarify that heterotypic collision experiments were meant not to further test the model but to describe differences with homotypic collisions and provide additional design options for tissue tessellations. Additionally, connected to our physical model for homotypic collisions, we provide new experiments using blebbistatin to probe the physical origin of the forces that drive boundary motion (Fig. S6; see lines 250-263). We believe that these revisions clarify the range of applicability of our model and provide additional insight into the driving mechanisms of boundary motion upon tissue collision.

Specific comments:

2.1 A major concern is the over-commitment to the model that the event is "solely based on cell density gradient", without considering the possibility that cell density gradient may be secondary as a downstream result of a primary mechanism.

Response: We agree that the phrasing "solely based on cell density gradient" was a poor choice of words and have revised that part of the abstract, which now reads "Next, we propose that genetically identical tissues displace each other based on pressure gradients, which are directly linked to gradients in cell density. We present...".

We agree with the reviewer that, in general, gradients in cell density can be the downstream result of some primary mechanism, and added this important point to the discussion section (lines 483-489). However, we studied collisions between tissues that were initially prepared to have different cell densities. Therefore, in this case, the cell density gradient did not result from any primary biological mechanism but was imposed experimentally. We also studied collisions between tissues that started expanding with the same cell density but different sizes. In this case, the tissues developed differences in cell density as they expanded, just as a result of the expansion kinematics (see our previous work in Ref. 10). In vivo, cell density gradients could potentially arise from a number of biological mechanisms. Uncovering such mechanisms and their consequences for tissue collisions in vivo is an interesting avenue for future research. In our paper, we propose that, however it arises, the cell density gradient drives boundary motion upon collision.

2.2. When addressing heterotypic dynamics where cell density failed to explain the observations, the authors then had to invoke cell-cell adhesion as the driving mechanism.

Response: We thank the reviewer for bringing up this point. We note that we did not try to explain the heterotypic collisions with the cell density model. In fact, we did not even measure cell density in heterotypic tissues. This is because heterotypic tissue collisions are more complex than homotypic ones, and border motion can be driven not only by cell density differences but also by differences in cell-cell adhesion and other biochemical signals. This is an important clarification, so we have articulated it more clearly in the revised manuscript (see lines 401-413).

Rather than using them to study the driving mechanisms of border motion, we use heterotypic collisions to report two key observations: (1) if one tissue is faster than the other, engulfment is possible, otherwise it cannot occur (Fig. 6g-j); and (2) mesenchymal cells, which have weaker cell-cell junctions, will tend to be severely displaced upon collision (Fig. 6a-d). These results thus emphasize that the cell density model, which applies to homotypic collisions, is insufficient for heterotypic collisions. Developing more complex models to describe the dynamics of heterotypic collisions is an interesting avenue for future research (lines 535-541). However, we do see our work as providing a framework for how to think about tissue collisions and as a demonstration of the kinds of tools and approaches that can be used to pattern complex composite tissues. To communicate these points more clearly, we have moved the section on heterotypic collisions to the end of the manuscript, where it intuitively follows that these results are not meant to test our physical model but rather to provide additional design options for tissue tessellations.

2.3 There were also difficulties in explaining the results with colonies of mismatched sizes based on cell density, where the argument that the smaller colony somehow had a low cell density also appeared unconvincing.

Response: Relative density at collision time, rather than at the time of stencil removal, determines the cell density gradient at the collision boundary. In Ref. 10 (Heinrich et al. eLife 2020), it was proven with experiments and simulations that larger epithelial tissues develop higher cell densities than smaller tissues as they expand due to relative rates of tissue spreading and cell division. Briefly, since both narrow and wide tissues expand with the same outer edge speed (see Ref. 10), smaller tissues will increase in *relative* area faster. For instance, tissue area for rectangular strips will double when the expanding edge has moved outward by half

of the tissue width, which is of course a smaller distance and reached more quickly in smaller tissues than larger tissues. This faster area increase means that, in the absence of cell division, smaller tissues will decrease in area due to spreading faster than larger tissues. Differences in cell division rates between smaller and larger tissues were not sufficient to compensate for this increased relative spreading of small tissues, so larger tissues indeed expand at higher density than smaller tissues, even when starting with the same density.

Since cell density kymographs are not the most direct way to visualize this effect (Fig. 2), we directly quantified the cell density of the wide and narrow tissues as they expanded, *before collision*, and plotted the results in new Supplementary Fig. 3. This plot clearly shows that, even though the two tissues begin with equal seeding densities, they develop different density profiles as they expand. The difference in density at collision then will clearly produce a density gradient.

2.4 A more plausible and unified hypothesis would treat cell-cell adhesion as the primary parameter, which may affect mechanical properties of the colony to determine the outcome for both homo- and heterotypic collisions; small, sparse colonies or weak cell-cell adhesions may lead to weaker integrated mechanical strength, against larger, denser colonies or strong cell-cell adhesions. The importance of cell-cell adhesion in tissue organization was in fact addressed decades ago in the elegant studies of Malcolm Steinberg (1930-2012, at Princeton!), which the authors should carefully consider.

Response: We thank the reviewer for this comment. As we discussed earlier, we focused on modeling the physics of homotypic collisions. For heterotypic collisions, we report experimental observations but do not attempt to provide a physical model. In homotypic collisions, we believe that cell-cell adhesion is not a primary determinant of collision dynamics because the two colliding tissues are genetically identical and cultured in the same way; therefore, they should have no difference in cell-cell adhesion. Hence, we built a model that can predict the collision dynamics even when the colliding tissues have the same cell-cell adhesion strength. To this end, here we propose that differences in pressure can drive the collision boundary. We propose that the differences in pressure arise from differences in cell density, which we observe in our experiments.

Even if it is not the primary parameter, cell-cell adhesion strength is still relevant for the dynamics of tissue collisions, even homotypic ones. For example, the amount of cell-cell adhesion will affect tissue stress as it controls the intercellular transmission of cell-generated contractility. We also agree that differences in cell-cell adhesion strength will likely be key for heterotypic collisions. Developing a physical model for

such collisions requires accounting for differences not only in cell-cell adhesion but also in traction forces, tissue stiffness, and cell polarity dynamics between the colliding tissues. Developing such a physical model is an interesting research direction beyond the scope of our manuscript.

Finally, Steinberg's differential adhesion ideas are certainly inspiring but not directly relevant for our work as we do not study tissue organization processes. In our experiments, tissues collide and displace each other but maintain their respective ordering in space. Moreover, as we discussed earlier, our physical model applies to homotypic collisions, in which the colliding tissues have equal cell-cell adhesion strengths.

2.5 The mechanical model as expressed in equation 1-3 has underlying, risky assumptions that must be spelled out and justified. The most concerning is that elastic modulus of a colony is a constant for a given cell type, while in fact it is likely a function of both cell density and cell-cell adhesion; weak or sparse cell-cell adhesion would likely lead to a low modulus and/or viscous behavior. Thus, the derivative of K with respect to ρ should not be treated as a constant in the equation for P' . This caveat casts doubt on the validity of the estimation of elastic modulus.

Response: We agree with the reviewer that the mechanical properties of tissues could in principle change with cell density. The mechanical response of a tissue to compression and extension is encoded into the so-called equation of state, which expresses the pressure as a function of the cell density. Establishing an equation of state for tissues remains a long-standing problem in the field of tissue mechanics. Not only could the stiffness modulus of a tissue be a function of cell density, but the actual functional form of the function $P(\rho)$ remains to be established. For example, to our knowledge, this function has not been experimentally measured yet. We do not attempt to do this in this work.

However, theoretical work by Recho et al. (Ref. 47) has proposed an equation of state based on principles of nonequilibrium thermodynamics. This equation of state is a logarithmic function of cell density, and it contains a prefactor with units of bulk modulus (Eq. 3). In this setting, the prefactor is just a number, not a function of cell density. The bulk modulus K as it appears in Eq. 3 is the coefficient that characterizes the linear mechanical response of the monolayer around the reference cell density ρ_e . We now state this explicitly in the revised manuscript.

Experimentally verifying whether Eq. 3 is an accurate equation of state for epithelial monolayers requires measuring pressure, which is beyond the scope of our work. Here, instead, we propose to use the theoretical prediction to infer the mechanical properties of cell monolayers from collision dynamics. Regardless of the accuracy of the equation of state, we believe that using collision dynamics to probe tissue mechanics is a valuable idea that we provide to the field.

Finally, we agree that a more complex formulation of K related to local density is desirable. However, this is a long-standing problem in the entire field of tissue mechanics and the necessary equations of state simply do not exist. Here, following Recho et al., we opted for the simplest parametrization rather than adding further assumptions to the model, and set K as a constant. Since these are important points, we now explicitly state that the chosen equation of state has yet to be verified by experiments, and that it might be generalized to contain a density-dependent bulk modulus in future work. (See lines 275-290).

2.6. It was proposed that monolayer tessellation can serve as an engineering tool. However, the application is premature due to the poor understanding of heterotypic collision, which is required for the construction of complex tissues.

Response: We have simplified and qualified the language about potential biomedical applications. We now discuss specific emerging applications for massive-scale tessellation of many smaller, homotypic epithelial monolayers in chronic wound treatment as a big-picture motivation (lines 49-51; 354-362). We also try to better emphasize the point that our findings and the predictive simulation tool we developed to computationally simulate and then build complex composite tissues may be useful in different spaces, including simple applications such as simply determining how to tile tissues in a Petri dish for maximum throughput (lines 505-534).

2.7. While the proposed process works for homotypic monolayers, the construct is no different from what can be obtained more easily by placing cells on micropatterned substrates.

Response: We agree that there are a range of techniques, and we have added further discussion of this in lines 354-362 and in our *Discussion* (lines 517-541), where we specifically highlight several recent and exciting patterning approaches. That said, we are unaware of any standard approaches that *easily* support > 2 types of populations (e.g. cell types, colors, etc.) *and* enable outgrowth (rather than imposing confinement); nor any with the speed, low-cost, accessibility, and scale of our

approach. While we know there are many clever patterning approaches in general, we are unaware of any demonstrations similar to the systematic assessment we provide in Fig. 1, nor the mosaics we construct later that play out at centimeter+ scale. Therefore, we believe that the patterning demonstrations that we present will be valuable tools for the broad community that studies tissue biophysics and engineering as well as for those many researchers who use cell monolayers as platforms for tissue biology studies.

2.8. It is unclear what exactly was learned from tri-tissue collisions. The "escape" may be easily explained when an expanding colony (purple cells in Fig. 6) encounters a barrier with a region of weak adhesion (where two green colonies meet in Fig. 6). Also, as a minor comment, the term and the figure are confusing in the sense that panel 6a shows 5 colonies with 2 colors, where the meaning of "tri"-tissue needs to be figured out between the lines.

Response: We have improved our discussion of the tri-tissue collisions (lines 366-375), explaining that the inner tissue migrates in between the outer tissues in a process similar to extrusion. The main point of this part of the manuscript is to highlight that the dynamics of collisions between three or more tissues do not simply reduce to those of tissue pairs. This is exemplified by escape events: the increased speed and escape of the central tissue does not have a counterpart in binary tissue collisions.

Regarding the explanation of the escape phenomenon, we are unsure what the reviewer is describing as "a barrier with a region of weak adhesion (where two green colonies meet)". We note that Fig. 5 (Fig. 6 in the previous version) shows collisions between identical tissues (only labeled with different colors). Therefore, the adhesion of green-green and green-purple cells is the same; there is no region of weak adhesion. While green-purple cell-cell adhesions are indeed less mature than green-green cell-cell adhesions, this speeding up of migration speed is not seen in the self-colliding tissue (Fig. 5h) where there are similar newly-formed cell-cell adhesions at the self-colliding tissue boundary. We also emphasize that escape events do not emerge from the purple tissue 'pushing' between the two green tissues. The 'escape' can only occur if there is an open path in front of the extruding tissue (purple in Fig. 5, compare Supplementary Videos 11 and 12). Hence, there is no "barrier". Finally, we believe that "tri-tissue collisions" is an appropriate description for the collisions in Fig. 5 because the escape process is a feature of ternary tissue collisions (as opposed to the binary tissue collisions studied in the first part of the paper). We added labels to Fig. 5 a and b, with descriptions in the caption to clarify this point.

2.9 (minor) Figure 2, it would be helpful to mark the points of colony fusion in panels e and f. Clarify if e and f are from the same set of data. Indicate the center of expansion in f.

Response: This is a good point and we have updated Fig. 2 and its caption accordingly.

2.10 In summary, the study starts with novel and very promising experiments, but the subsequent sections scattered into various territories with significant caveats. In addition, the sections on tessellation and cell escape, which serves as no more than shiny objects at this stage, should be removed.

Response: While we understand the previous storyline was somewhat fragmented and confusing, we respectfully disagree with Reviewer 2 on the issue of 'shiny objects'. To better address this, we have worked to restructure the paper to tell a more cohesive narrative. We have also extended our discussion on the context of both tessellations and the escape events to better emphasize their importance. Briefly, we feel that both the tessellation demonstration and the escape phenomenon highlight previously unreported and surprising dynamics of multi-tissue interactions of relevance to a wide range of communities. We have provided additional context below to explain how we have clarified the impact of these findings in the revised manuscript.

Whereas previous studies had only studied head-on binary collisions between rectangular tissues, the tessellations allowed us to study collisions between many tissues at arbitrary angles—an advance of significant biophysical interest. This opens the door to characterizing and understanding the dynamics of multi-tissue collisions in future works. The tessellations further demonstrate that tissue collisions are very reproducible up to macroscopic scales, and how collision dynamics can be harnessed for design purposes. For instance, we have added further discussion in the text about recent biomedical technologies specifically based around massive tiling of human epidermal patches that are being used to treat chronic wounds. Our 'Escher' figure is a reduction to practice of the implication that such large-scale tessellations can be predicted and realized experimentally, and that the resulting tissues maintain integrity during a cell sheet transfer process, which was unexpected to us.

The escape phenomenon is also quite surprising as there is no prior data on tri-tissue interactions, and it was not obvious that the cell speed-up and escape process

we observed could occur. The fact that such narrow, nearly single-cell-wide strands of one tissue can co-exist within another and be generated through a collision process is unexpected and interesting. Please see our response to point 2.8, where we detail the improvements we have made to the escape discussion.

Finally, our simulation approach should eventually be generalizable to heterotypic interactions as well, assuming a similar characterization and modeling approach is performed for each unique tissue-tissue interaction. We have moved the section on heterotypic collisions to the end of the paper, and we now highlight it as an area for continued, future work.

Reviewer #3 (Remarks to the Author):

Heinrich and co-workers studied the dynamics of collision of multicellular tissues. They combined experiments and modeling to predict how the interface between two colliding tissues depends on tissue shape and cell density density. They simulated the expected position of the collision front for various combinations of rectangles and circles using a previously published method and showed that the predicted front fit well the experimental findings. The main result is that the collision front does not move once the cells collide. By looking at genetically identical tissues that varied in size and density, they showed that density gradient is the main control parameter over the motion of the collision front. They modeled the tissue as a dry active compressible elastic material. They recovered that the collision boundary moves from high to low densities with a speed that is proportional to the density gradient.

Next, they studied heterotypic tissue collision. They used three breast cancer cell lines: MCF10A (benign), MCF7 (malignant, non invasive) and MDA-MB-231 (metastatic) cells. They showed that higher expansion speed does not imply higher collision strength.

Finally, the authors develop a new design tool named TissElate, which allows them to design complex tissue tessellation based on their initial model for tissue expansion. They showed that different initial seeding geometry leads to predictive self-assembled tissue tessellations. They showed that tri-tissue collision can lead to escape events, where cells within the escape region can move faster than any other cells in the expanding front.

I think this is an elegant paper that presents novel and robust results. The results and the software developed would be of interest for physicists, biologists and bioengineers interested in tissue mechanics and collective cell migration. The manuscript was clear and well organized. I

just have a few comments and suggestions below that might strengthen the paper, in particular the comparison between experiments and theory.

Major comments:

3.1. Justification of the model assumptions could be improved.

- Why is the friction independent of density?

Response: We agree with the reviewer that the friction coefficient should in principle depend on cell density. We assume that it does not for simplicity. We note that the relevant density dependence in the model is that of the pressure, which is the driving force of cell motion. While a density-dependent friction coefficient would affect the speed of the boundary, it cannot change boundary motion direction because friction does not drive motion. The direction of the boundary motion is determined by the density dependence of the pressure. We now explain this in the manuscript (see lines 184; 214-218).

- Experimental justification for choosing the specific equation of state beside theoretical work by Recho et al.

Response: To our knowledge, there are no experimental measurements of equations of state for tissues, let alone to the level required to distinguish between different models. Therefore, there is no experimental support for one specific equation of state. Respectively, there are only a few theoretical proposals for tissue equations of state. We chose the one by Recho et al. because it is derived using the principles of nonequilibrium thermodynamics and it applies to viscoelastic cell monolayers with cell proliferation around the homeostatic state. These are quite general conditions, which we believe are met in our experiments; hence our choice. We now explain this in the manuscript (see lines 280-290).

- Why doesn't K depend on cell density?

Response: This question was also raised by Reviewer #2. Please see our reply to point 2.6 above. Briefly, the equation of state in Eq. 3 has a logarithmic dependence on cell density, and it contains a prefactor K with units of bulk modulus. In this setting, the prefactor is just a number, not a function of cell density. In the derivation by Recho et al., the bulk modulus arises from one of the kinetic coefficients of the Onsager flux-force relations used in the nonequilibrium thermodynamics formalism. Hence, it acts as a constant that characterizes the linear response of the system to bulk deformations around the reference density ρ_e . Thus, all the density dependence goes into the logarithmic function that follows it. Generalizing the equation of state

to include a density-dependent bulk modulus is an open question and an interesting direction for future work. We now explain this in the revised manuscript.

- How does the density at the collision front depend on the initial seeding densities?

The entire density field in an expanding tissue depends both on its initial density and size. We studied the size dependence in our previous work (Ref. 10). In particular, we found that initially-larger tissues develop denser cores. Here, we found that initially-denser tissues remain denser during expansion through the onset of the collision (Fig. 2f, right). We now note this explicitly in the revised manuscript (lines 197-203) and have prepared an additional supplemental figure directly showing that the density gradient at the collision time correlates with the initial density mismatch (Fig. S4).

3.2. The comparison between the model and the experiments could be strengthened. At the moment, the authors only show that velocity and cell density gradient are negatively correlated. While they derive an equation of state for the tissue, I have not seen a direct comparison between the experimental plots (Fig. 2H) and the equation of state (equations 2 and 3). More quantitative comparison would strengthen the claim that mechanical properties of a mono layer can be infer by studying the kinematic of the colliding fronts.

Response: We thank the reviewer for raising this point. We agree that a more accurate comparison would be desirable. However, experimentally verifying the equation of state requires measuring not only the cell density but also the pressure, which is beyond the scope of this manuscript. Thus, our approach is to take an equation of state that has been derived theoretically for cell monolayers like those in our experiments, and to employ it to rationalize our data and extract parameter values. Even though we cannot provide direct experimental validation of the equation of state that we use, we find it instructive to use it to predict the velocity of collision boundaries. As we show in Fig. 2H, our predictions can fit the data, from which we extract a sensible value of the bulk modulus of the tissue. We hope that this proof of concept motivates further work in both measuring equations of state for tissues and using collisions to probe tissue mechanics. We hope that this future work will enable more accurate, quantitative comparisons like the ones envisioned by the reviewer.

Suggested experiments to strengthen the comparison between experiments and theory:
A) increase the density mismatch between the two colliding tissues (right now it is only 1.5X) this parameter could be explored experimentally.

B) Does the absolute density also impact the mechanical properties of the tissue? To test that, maybe the authors could explore the collisions of two tissues with the same density gradients as before, but with a different cell densities on both sides.

Response: These are good suggestions, so we have performed additional experiments to address them:

- A) We increased the initial density mismatch. Whereas it was in the range 1.3-1.5 in our original experiments, we have now reached much higher initial density ratios of up to 2.6. Higher initial density mismatches lead to higher density gradients upon collision, and hence to faster boundary motion, which provides further support for our hypothesis. We have added the new data to Fig. 2H and provided additional plots in Fig. S5a-b. We also revised our estimate of the bulk modulus by fitting our model to the new, extended data set.
- B) We also tested the role of the absolute cell density, which we varied from around 1200 cells/mm² to around 3300 cells/mm². We found that the absolute cell density does not significantly affect boundary motion, again consistent with our hypothesis that cell density gradients drive boundary motion. We show the corresponding data in the new Fig. S5c.

We discuss these new results in a new paragraph in the section “Cell density gradients drive boundary motion” (see lines 229-246).

3.3. Could the authors comment on why they think there is a 2 orders of magnitude difference between the stiffness under compression and the stiffness under extension?

Response: The mechanical response of tissues is likely to be highly nonlinear. Hence, the response to compression and to extension need not be the same. For example, both the cytoskeleton and cell-cell adhesion might stiffen under extension but not under compression. In general, the cellular mechanisms responsible for extensional and compressional tissue elasticity can be quite different, and hence the corresponding moduli may differ too (see added Ref. 51 for a more complete discussion).

The values for stiffnesses provided in the text were drawn from two radically different studies: one that probed the tensile elasticity of freely suspended epithelia (no ECM or substrate at all, Ref. 48), and another that studied the effective elasticity of a monolayer spreading by collective cell migration on a substrate (Ref. 49). These are the best values that we could find, which highlights the need for further, more standardized experimental measurements.

Our estimate corresponds to yet a different measurement: compression upon tissue collision. However, we note that our revised estimate does not have a 2 orders of magnitude difference with the others. Rather, our estimate is 10 times smaller than the tensile stiffness of suspended monolayers (Ref. 49), and 4 times larger than the effective tensile stiffness of expanding monolayers. To clarify this comparison, we have extended the discussion and added a reference (Ref. 51) to better explain the expected differences between extensional and compressional moduli of cell monolayers (see lines 307-315).

3. The connection between the results from the homotypic collision experiments + model with the heterotypic collision experiments is a bit weak. This section is disconnected from the others. Can the model derived for MDCK cells help identify what physical properties of the homotypic tissues drives the motility of the colliding front?

Response: We agree that the heterotypic tissue collisions could have been more clearly presented. Our main goal was to first study homotypic tissues and then use them to design tessellations. We included the heterotypic collision experiments primarily to emphasize that the simple physical principles that we found for homotypic collisions need to be generalized in the future to explain heterotypic collisions. In particular, whereas we find that cell density gradients are the main driver of boundary motion in homotypic collisions, other ingredients like differences in cell-cell adhesion, traction forces, contractility, and tissue stiffness will also contribute in heterotypic collisions. These differences are not included in our physical model, and hence we did not use it to interpret the observations of our heterotypic collisions. To clarify this, we have moved the section on heterotypic collisions to the end of the manuscript, after discussing tessellations. This clarifies that we do not attempt to use heterotypic collisions to further test our physical model, but rather to showcase additional design options for tissue tessellations. Generalizing the model to account also for heterotypic collisions is an interesting direction for future work. In the revised manuscript, we have made these points explicitly to avoid confusion (see lines 398-413).

Minor comments:

1. The caption for figure 3D says the picture was taken 24h after removing the stencil while the text in the figure says 36h.

Response: The caption says 24 hours after the collision. The collision occurs at $t=12$ h, so 24 hours after the collision corresponds to $t = 36$ h. We have now clarified this in the caption.

2. There is no mention of what cell types are/can be modeled with the Tissellate software. I assumed MDCK cells with identical seeding densities. Can this be extended to other cell types? To heterotypic collision?

Response: Tissellate is a generic software that can model the expansion and tessellations of tissues. Since the algorithm is only based on the kinematics of tissue expansion and the dynamics of tissue boundaries, it is not specific to a cell type. Any cell type that forms confluent monolayers in culture would be appropriate for modeling with Tissellate. We used the software on MCDK monolayers with equal seeding densities. However, using it on monolayers with different densities should only amount to changing the kinematic parameters of tissue expansion. Even so, this would only necessitate a parameter change when resolution under 200 μm is required, because the tissue boundary stops after translating roughly this distance. Generalizing the algorithm to heterotypic tissue collisions might require introducing new rules for the behavior of collision boundaries, which is an interesting direction for future work. We now explicitly mention this point at the end of the discussion.

REVIEWERS' COMMENTS

Reviewer #2 (Remarks to the Author):

I read the responses from the authors then the revised manuscript with a fresh pair of eyes. While the better separation of homo- and hetero-typic experiments and the addition of blebbistatin experiments represent improvements, I still feel that the manuscript is unnecessarily long, with some experiments serving as little more than fancy distractions. A shorter, better-focused paper will likely be stronger in both quality and impact. In addition, the paper can be strengthened by stating the exact significance of each experiment; what specifically does it teach us about biology or lend itself to biomedical applications. As an example, it is difficult to appreciate the importance of the tri-tissue experiment by reading broad-stroke sentences like "These findings suggest that escapes are an emergent dynamical property of three-tissue interactions" (and equally vague statements added to Discussion).

The basic finding, that boundary movement between homotypic cell colonies follow definable rules, remains innovative and attractive. However, the study can go deeper into the underlying mechanisms rather than move around scattered topics. Here I do not mean identifying molecular interactions but pinpointing the key driver of the phenomenon. The authors noticed a difference in cell density then plunged single-mindedly into the idea of cell density being the determinant, while in reality the most important factor may be something else, for example cell-cell adhesions that may be affected by cell surface-volume ratio thereby cell size and density.

The section of modeling remains unnecessarily long, with some notable weaknesses as pointed out previously. I shall in addition point out that the authors first assumed "that the tissue pressure P increases with cell density ρ ", then in the next sentence predicted that cell "boundary moves from high to low cell densities with a speed proportional to the density". This goes little beyond saying that higher pressure makes cell move faster, which hardly represents a fresh insight.

It is mystifying why the artwork of M.C. Escher is relevant to the present study, as I cannot see any connection between the masterful optical illusion of Escher and tissue repair. How about other more down-to-earth patterns that better reflect the clinical relevance, such as something more relevant to "CelluTome"?

The section on heterotypic tissue boundary dynamics remains a confusing distraction, given its possibly different mechanism from homotypic dynamics as the authors contend. However, it would make more sense if the authors were able to show differential adhesions as the common driver for both processes as eluded to above. Otherwise, it would be better to mention heterotypic dynamics only briefly in Discussion as future work to avoid misleading readers.

Reviewer #3 (Remarks to the Author):

I would like to thank the authors for their response and the updated manuscript. I believe the modifications greatly improve their paper, in particular regarding the comparison between experiments and theory as suggested by two of the reviewers. With the new experiments and the rewriting, this manuscript is now ready for publication in Nature Communication.

REVIEWERS' COMMENTS

We appreciate the editorial and reviewer comments and have addressed all comments both in this document and with changes to the main text. Previous revisions have been converted to 'black' and new revisions in the main text are highlighted in blue.

Reviewer #2 (Remarks to the Author):

1. I read the responses from the authors then the revised manuscript with a fresh pair of eyes. While the better separation of homo- and hetero-typic experiments and the addition of blebbistatin experiments represent improvements, I still feel that the manuscript is unnecessarily long, with some experiments serving as little more than fancy distractions. A shorter, better-focused paper will likely be stronger in both quality and impact. In addition, the paper can be strengthened by stating the exact significance of each experiment; what specifically does it teach us about biology or lend itself to biomedical applications. As an example, it is difficult to appreciate the importance of the tri-tissue experiment by reading broad-stroke sentences like "These findings suggest that escapes are an emergent dynamical property of three-tissue interactions" (and equally vague statements added to Discussion).

Response: We thank the reviewer for encouraging us to clarify the significance of each experiment. We have now done so throughout the text. Specifically, we have added sentences to clarify the specific goals of each of our four experiments. These sentences appear at the beginning of sections "Collisions between archetypal tissue pairs," "Homotypic tissue boundary dynamics and collision memory," "Large-scale tissue tessellations for cell sheet engineering," and "Heterotypic tissue boundary dynamics." We believe that this clarifies what we set out to learn with each experiment.

Furthermore, we have also clarified the conclusions of our tri-tissue collision experiments. In the revised manuscript, we explain more clearly that escape events during tri-tissue collisions are surprising phenomena that are not a direct consequence of the resulting multi-tissue geometry; they cannot be explained based on the dynamics of a single tissue with the geometry of the colliding tissues. Rather, escapes result from the interactions between colliding tissues, which depend on their initial conditions and expansion history before the collision. Moreover, escapes are not predicted by our simulations based on binary collision rules. Therefore, escapes are a multicellular process that emerges from the simultaneous interactions of at least three tissues..

2. The basic finding, that boundary movement between homotypic cell colonies follow definable rules, remains innovative and attractive. However, the study can go deeper into the underlying mechanisms rather than move around scattered topics. Here I do not mean identifying molecular interactions but pinpointing the key driver of the phenomenon. The authors noticed a difference in cell density then plunged single-mindedly into the idea of cell density being the determinant, while in reality the

most important factor may be something else, for example cell-cell adhesions that may be affected by cell surface-volume ratio thereby cell size and density.

Response: We agree with the reviewer that cell density could potentially affect cell-cell adhesion strength. An adhesion difference could then imply differences in tissue-scale forces that could drive boundary displacement upon collision. This mechanism is implicitly included in our model, where a difference in cell density produces a pressure difference that drives boundary displacement. As suggested by the reviewer, differences in cell-cell adhesion could be involved in translating differences in cell density into differences in pressure. As we now discuss in the manuscript, we defer this point to future work. In any case, our model provides a way to predict boundary motion based on the control parameter of our experiments, which is the cell density of the colliding tissues.

We also emphasize that our findings cannot be explained solely by differential adhesion. While differential adhesion is known to drive cell sorting, in our experiments cells are already sorted from the start. In the classical Steinberg framework of differential adhesion, the tissue would remain in the cell-sorted steady state without boundary motion. Here, instead, we found that the tissue boundary moves even when cells are sorted and, in fact, even for genetically-identical cells. Hence, we can't rely on cell-cell adhesion differences alone to explain boundary motion in collisions between homotypic tissues. We observed that boundary motion occurred whenever there was a difference in cell density, and we found that the boundary displacement was faster at higher cell density differences. Hence, we chose to build a model that, based on established physical principles, captures boundary motion driven by cell-density gradients. In future work, it will be interesting to see if and how differences in cell density lead to differences in cell-cell adhesion.

3. The section of modeling remains unnecessarily long, with some notable weaknesses as pointed out previously. I shall in addition point out that the authors first assumed **"that the tissue pressure P increases with cell density ρ "**, then in the next sentence predicted that cell **"boundary moves from high to low cell densities with a speed proportional to the density"**. This goes little beyond saying that higher pressure makes cell move faster, which hardly represents a fresh insight.

*Response: We respectfully disagree with the referee. First, the quote by the referee misses the last word of the sentence, which is crucial. Our second sentence reads that Eq. 2 "predicts that the collision boundary moves from high to low cell density with a speed proportional to the density **gradient**." Second, within our model, boundary motion is not a result of cells moving faster at higher pressures. Rather, through interactions like contact inhibition, epithelial cells tend to move more slowly (not faster) at higher densities, and hence at higher pressures. Our model predictions are not based on these effects. In fact, our model does not incorporate autonomous cell motility. Instead, our model takes into account that higher cell densities yield higher pressure in the cell monolayer. Then, pressure gradients drive flows, which translates into boundary displacement in our case. Finally, given these nuances in the modeling discussion, we feel that the length of the modeling section is appropriate.*

4. It is mystifying why the artwork of M.C. Escher is relevant to the present study, as I cannot see any connection between the masterful optical illusion of Escher and tissue repair. How about other more down-to-earth patterns that better reflect the clinical relevance, such as something more relevant to "CelluTome"?

Response: Our work shows that collisions between multiple epithelia can produce tissue tessellations. Moreover, we emphasize that the resulting tessellations can be rationally designed via simulation. We presented several examples of complex tissue tessellations, one of which was inspired by Escher's work on rhombille tiling (e.g. his Metamorphosis series). We emphasize that Escher's work does not have any direct connection to our scientific results; rather, it provides inspiration for the complex tessellations that we demonstrate.

While we have also validated a CelluTome-style geometry (simple array of circles that tessellate into a checkerboard), we chose to report the more complex rhombille tiling to better communicate that arbitrary tilings can be designed using our approach. Finally, using Escher-inspired tilings or other patterns does not affect the validity of our work and conclusions. As some of our work was directly inspired by Escher's art, we chose to include these patterns and to properly credit Escher as the inspiration.

5. The section on heterotypic tissue boundary dynamics remains a confusing distraction, given its possibly different mechanism from homotypic dynamics as the authors contend. However, it would make more sense if the authors were able to show differential adhesions as the common driver for both processes as eluded to above. Otherwise, it would be better to mention heterotypic dynamics only briefly in Discussion as future work to avoid misleading readers.

Response: As we discussed in our response to point 2, differences in cell-cell adhesion alone cannot explain our observations of boundary motion in homotypic tissue collisions. Cell-cell adhesion is most likely not the only factor driving boundary motion in heterotypic collisions either. Therefore, we cannot unfortunately not show that cell-cell adhesions are the sole and common driver of all the collisions.

In the previous revision, we reformatted the manuscript to clearly separate heterotypic from homotypic collisions. Moreover, we explained in lines 415-427 that we did not use our model to describe heterotypic collisions. Yet, our data on heterotypic collisions shows how they can be used to engineer different tissue tessellations (e.g. involving engulfments) beyond those possible in homotypic collisions. We clearly explained this rationale in the heterotypic collisions section to avoid misleading readers. Finally, we find that the dynamics of heterotypic boundary motion are distinct from those of homotypic collisions. We believe that it is important to show these results (as a result, not just a discussion), and that they will be useful for future studies. While it would indeed be convenient if differential adhesion could explain all of our data, we direct the reviewer to our previous response in our earlier rebuttal where we discussed this. While there is a connection, which we discuss, between cell-cell adhesion and cell density,

classical differential adhesion as originally proposed by Steinberg does not seem sufficient to explain our data as we discussed both in our previous rebuttal round and in point 2 here.

Reviewer #3 (Remarks to the Author):

I would like to thank the authors for their response and the updated manuscript. I believe the modifications greatly improve their paper, in particular regarding the comparison between experiments and theory as suggested by two of the reviewers. With the new experiments and the rewriting, this manuscript is now ready for publication in Nature Communication.

We thank the reviewer for their positive assessment of our revisions.